# Profile Evolution and Cross-Process Collaboration Strategy of Bearing Raceway by Centerless Grinding and Electrochemical Mechanical Machining

**DOI:** 10.3390/mi14010063

**Published:** 2022-12-26

**Authors:** Zhaobin Yan, Shuangjiao Fan, Wenpeng Xu, Zhixin Zhang, Guibing Pang

**Affiliations:** College of Mechanical Engineering and Automation, Dalian Polytechnic University, Dalian 116034, China

**Keywords:** rotary parts, bearing raceway, electrochemical mechanical machining, centerless grinding, surface profile, roundness

## Abstract

Roundness is one of the most important evaluation indexes of rotary parts. The formation and change of roundness in the machining of parts is essentially the formation and genetic process of profile. Centerless positioning machining is one of the main surface finishing methods of rotary parts. The rounding mechanism of centerless positioning machining determines its unique roundness profile formation and genetic characteristics. How to eliminate the roundness error of centerless positioning machining has become one of the important issues in the research of high-precision rotary part machining. This paper explores the influence of process parameters on the roundness error from the perspective of profile evolution during centerless grinding and electrochemical mechanical machining, with the aim of providing a cross-process collaboration strategy for improving bearing raceway accuracy. Through an experiment of centerless grinding, the influence law and mechanism of process parameters on the profile are discussed. On this basis, electrochemical mechanical machining experiments are designed to explore the variation rules and mechanisms of different profile shapes in the machining process. The cross-process collaboration strategy is studied, and reasonable parameters of centerless grinding and electrochemical mechanical machining are determined. The results show that in the centerless grinding stage, increasing the support plate angle can form a multiple-lobe profile with high frequency within a wide range of process parameters. Electrochemical mechanical machining can effectively smooth the high-frequency profile and appropriately expanding the cathode coverage can improve the roundness error and reduce the requirement of initial accuracy of a multiple-lobe profile workpiece to a certain extent. Therefore, the combined machining technology of “centerless grinding + electrochemical mechanical machining” provides an efficient technical means to realize the precision machining of rotary parts such as bearing raceways.

## 1. Introduction

High-precision rotary parts are indispensable in modern industry [1,2]. For example, high-precision bearings are widely used in aviation, railroads, automobiles and other industries. High-precision bearings are more expensive than ordinary bearings because of low processing efficiency, harsh process conditions and low yield [3,4]. Therefore, the research of advanced machining technology for precision rotary parts has important theoretical and application value. Roundness is one of the most important accuracy evaluation indexes of rotary parts, and the roundness profile is the graphical basis of roundness error. The formation and change of roundness error in parts processing are essentially the formation and genetic process of the roundness profile [5,6]. Centerless positioning machining is one of the main ways to finish the surface of rotary parts, and it is an irreplaceable process in the machining of rotary parts such as bearing raceways, bearing rollers and precision shafts [7,8,9].

The rounding mechanism of centerless positioning processing determines its unique roundness profile formation and genetic characteristics. How to eliminate the roundness error of centerless positioning has become one of the important issues in the field of high-precision rotary parts manufacturing [10,11,12]. Stancekova et al. [13] analyzed the influences of revolution of the feed roll, grinding wheel speed and height of the guide bar parameters on cylindricity deviation during centerless grinding of 50CrMo4 steel shaft. Bianchi et al. [14] considered the influence of nonlinear separation of wheel-workpiece detachment under three-lobe profiles on the stability of the centerless grinding process. Wu et al. [15] analyzed the relationship between the waviness decrease rate and the dynamic components of the grinding force in the centerless grinding process and found that the frequency characteristics of the waviness decrease rate showed a similar tendency to those of the grinding force. Klocke et al. [16] found that the interaction between the cutting process and the main resonance of the machine structure would cause the workpiece center to oscillate, resulting in irregular material removal, thus increasing the workpiece waviness. Epureanu et al. [17] proved that the instability of a centerless grinding system was inevitable, but the amplitude of waves generated on the machined surface had an upper limit, and the only possibility to improve the machined surface was to generate a polygonal cross-section profile. Cui et al. [18] created a three-dimensional simulation model of through feed centerless grinding processes based on homogeneous coordinate transformation and used it to study the influence of workpiece geometric shape change and regulating wheel motion state on workpiece roundness error. The problem of centerless positioning processing is that the processing parameters, workpiece center displacement, and the original error of the processing surface all affect the stability of the process system to a certain extent and then affect the processing results. Since the error form of the blank after heat treatment is random, the workpiece center will also shift with the processing. When the process system is adjusted to correct the raceway profile error of a certain form or the center position of a workpiece, the correction may not be valid for workpiece of other error forms and the center positions. Therefore, it is difficult to consistently obtain good raceway accuracy with centerless grinding or centerless superfinishing.

Aiming at the problem that the centerless positioning machining cannot fundamentally improve the roundness of rotary parts, this paper applies electrochemical mechanical machining to the subsequent machining of centerless grinding. Electrochemical mechanical machining combines the electrochemical action and mechanical action, which can efficiently polish the surface of parts and improve the machining accuracy of parts to a certain extent [19,20]. Grabowski et al. [21] found that electrochemical assistance can effectively improve the surface quality in the machining process by comparing the surface quality and precision of 1.4301 stainless steel after turning with or without electrochemical assistance. Ye et al. [22] used a wedged end tube tool for electrochemical milling the deep and narrow groove of GH4169 nickel-based alloy and explored the wedge angle of the cutting tool to improve the electrolyte flowed field and machining quality through simulation. Liu et al. [23,24] explored the machining characteristics of micro-milling with a high-speed spiral electrode of nickel-based superalloy 718. The electrochemical micro-milling model with high-speed rotating was established based on the finite element analysis method, and the change of workpiece surface profile was predicted by the simulation of the machining electric field. Yang et al. [25] conducted theoretical and experimental research on the mechanism of laser and shaped-tube electrochemical milling (Laser-STEM). Through the experiment, it was found that the efficiency and precision of the special section tube micro-groove machining could be improved by using synchronous laser-assisted STEM. Huang et al. [26] proposed the ultrasonic-assisted electrochemical drill-grinding technology to reduce the surface roughness of the hole sidewall of the nickel-based superalloys from Ra 0.99 to Ra 0.14 µm. The existing research found that the electrochemical mechanical machining process could not only ensure the machining accuracy but also improved the harmonic profile with high frequency. Wang et al. [27] used the electrochemical mechanical polishing technology under the optimal parameter combination to process the GCr15SiMn rolling bearing raceway. The results showed that the surface roughness, profile roundness and waveness were greatly improved. Wei et al. [28] proved through the electrochemical mechanical polishing experiment of stainless steel rod that the processing technology can effectively improve the surface waveness of the workpiece. Pathak et al. [29] used pulse electrochemical honing for straight bevel gears made of 20 MnCr5 alloy steel, which greatly improved the surface waveness while improving the surface roughness. After processing, the average waviness of the original workpiece was reduced from 5.04 to 1.58 μm, and the maximum waviness was reduced from 30.17 to 5.52 μm.

Aiming at the problem that centerless positioning machining cannot fundamentally improve the roundness of rotary parts, this paper proposes a cross-process collaborative optimization machining scheme that applies electrochemical mechanical machining to the subsequent machining of centerless grinding. Taking a bearing ring as the research object, a combined machining scheme of “centerless grinding + electrochemical machinery” for precision bearing raceway is proposed. From the perspective of roundness profile change, the influence law of process parameters on roundness profile shape is discussed. Firstly, centerless grinding experiments are designed to study the influence of process parameters on the profile shape and obtain the configuration rules of process parameters that can form different profile shapes. On this basis, electrochemical mechanical machining experiments are designed to explore the changing rules of different profile shapes and machining effects. Thus, the reasonable process parameter configuration principle and optimization range of centerless grinding and electrochemical mechanical machining are determined, and the cooperation of the two is realized, which provides a means for obtaining the cross-process parameter configuration of high-precision bearing raceways.

## 2. Influence of Centerless Grinding Process Parameters on Profile

### 2.1. Experimental Design

The experimental device is shown in Figure 1. The counterclockwise rotation of the grinding wheel is the main movement; the plate and the regulating wheel are the positioning mechanism of the workpiece, and the regulating wheel drives the workpiece to rotate clockwise. The workpiece surface is ground by the radial feed of the grinding wheel, and the regulating wheel is tilted horizontally to ensure the axial movement of the workpiece.

The geometric layout of the centerless grinding process system is the main factor affecting the harmonic profile of parts [30,31]. Previous studies have shown that the support plate angle and the center height are the main factors affecting the roundness profile and the error value. The center height refers to the distance from the center of the workpiece to the line between the center of the grinding wheel and the regulating wheel, and the support plate angle is the angle between the supporting surface of the plate and the vertical direction, as shown in Figure 2. Some research has shown that the harmonic height and wavelength of the roundness profile change with the above two factors [32,33,34]. Therefore, this paper designed the machining experiments with different support plate angles and different center heights, and the values of the parameters are shown in Table 1. The other experimental conditions are shown in Table 2. The particle size of abrasive is 80# in Table 2, which was proved to be an optimization parameter for workpiece surface quality in our previous research. For other process parameters in Table 2, as they do not play major roles in the surface profile shape, these parameters are not discussed, and general values are adopted in this study.

### 2.2. Experimental Results and Analysis

#### 2.2.1. Analysis of the Influence of Center Height and Support Plate Angle on Accuracy

Under the conditions of nine combination configurations of three kinds of support plate angles and three kinds of center height, ten workpieces were processed, respectively, and the roundness error value after processing was measured. Each piece was measured three times, and the average value was taken. The results are shown in Table 3. For the three center heights, when the support plate angle is 60°, the average roundness error after machining is minimal.

The interactive two-factor ANOVA was conducted on the data in Table 3, and the results are shown in Table 4, with a significance level of 0.05. It can be seen from Table 4 that under the combination of different parameter configurations set in the experiment, the effect of center height on the roundness of the workpiece is not significant, while the effect of the support plate angle on the roundness of the workpiece is relatively significant, and there is no obvious interaction relationship between the two.

The results shown in Table 3 and Table 4 relate to the friction force generated between the regulating wheel and the workpiece. Figure 3 shows the effect of the support plate angle on the friction between the regulating wheel and the workpiece. When centerless grinding is used to process the outer circular surface of the workpiece, the extension line of the plate surface and the tangent line of the workpiece and the regulating wheel intersect at a V-shaped angle φ. When the support plate angle is too large, the V-shaped angle φ degree is large, the friction force between the regulating wheel and the workpiece is reduced, and the workpiece cannot form a smooth movement in the grinding area, resulting in large profile fluctuations and increased roundness error value. When the support plate angle is appropriately reduced, it is conducive to increasing the friction between the regulating wheel and the workpiece, and when the workpiece is separated from the grinding wheel, it can still be driven by the regulating wheel, and the grinding effect is strengthened. However, when the support plate angle is too low, the V-shaped angle φ degree is small, and the friction between the regulating wheel and the workpiece is too large, which will cause the vibration of the workpiece in the grinding area and can also cause the roundness error value to increase.

#### 2.2.2. Effect of Center Height and Support Plate Angle on Profile Morphology

Table 5 shows the morphological changes of workpiece profiles before and after machining under the conditions of different support plate angle and different center height combinations. It can be found that when the original profile of the workpiece is a three-lobe profile, when the support plate angles are 50 and 60°, and the center heights are 26, 32 and 40 mm, the processed profiles are multiple-edge profile, three-lobe profile and multiple-lobe profile. This indicates that the center height has a significant effect on the profile shape in a certain range of the support plate angle, and increasing or lowering the center height will make the profile a multiple-lobe profile. When the support plate angle is 70°, the processed profile shows a multiple-lobe profile. 

The reason for the change of profile is related to whether the workpiece moves smoothly during the grinding process, and the smooth movement of the workpiece is conducive to the formation of a multiple-lobe profile. As shown in Figure 4, for a certain support plate angle value, when the center of the workpiece is lower than the center line of the grinding wheel and the regulating wheel, the friction between the workpiece and the grinding wheel, and the friction between the regulating wheel and the support plate are relatively large, which is conducive to the stability of the workpiece movement, and it tends to form multiple-lobe profiles. When the center height increases, the friction between the workpiece and the grinding wheel, and the friction between the regulating wheel and the support plate decrease, the machining process has the tendency to be unstable, and it tends to form three-lobe profiles. However, when the center of the workpiece is higher than the center line of the grinding wheel and the regulating wheel, it is also conducive to the formation of a multiple-lobe profile, which may be related to the influence of the direction of friction on the stability of the process system, which is also a problem worthy of further study.

When the support plate angle is large, the distance between the workpiece and the tangent points of the grinding wheel, the regulating wheel and the support plate is more balanced, which is conducive to stable processing and obtaining multiple-lobe profiles.

### 2.3. Centerless Grinding Parameter Configurations with Different Precisions and Profile Shapes

According to the above analysis, in the centerless grinding stage, adjusting the support plate angle will significantly affect the roundness error of the workpiece. Under the three conditions given in this paper, higher roundness accuracy can be obtained when the support plate angle is 60°. Different combinations of the center height and the support plate angle can also significantly change the profile shape of the rotary surface. Appropriately increasing or decreasing the center height and increasing the support plate angle are conducive to the evolution of the profile into a multiple-lobe profile shape. The influences of the center height and the support plate angle on the roundness value and profile shape are shown in Table 6 and Figure 5.

## 3. Influence of Electrochemical Mechanical Machining Parameters on Profile

### 3.1. Experimental Design

It has been shown that electrochemical mechanical machining accuracy is mainly affected by electrochemical parameters [34,35]. Among the electrochemical parameters, the coverage of electrochemical action is the main influence on the profile. Therefore, two experimental studies are carried out in this paper: (1) The influence of electrochemical range on roundness is analyzed by adjusting the cathode coverage. (2) Specimens with different roundness values and profile shapes are processed by electrochemical mechanical machining to obtain the mechanism of their action on profile evolution. In this way, the reasonable electrochemical mechanical machining parameters are explored and the reasonable early centerless grinding process conditions are selected according to the influence law of the centerless grinding process results on the final machining results.

The experimental device is shown in Figure 6. The workpiece is connected to the positive electrode of the power supply, and the tool is connected to the negative electrode of the power supply. The electrolyte is passed through the gap between the workpiece and the cathode. The abrasive is pressed on the workpiece surface by cylinder pressure. When the machine spindle drives the workpiece to rotate, the alternating action of electrochemical action and mechanical action is generated on the workpiece surface to realize the workpiece surface machining. The main experimental parameters are shown in Table 7, and two different cathode coverage ranges are chosen as shown in Figure 7.

### 3.2. Experimental Results and Analysis

#### 3.2.1. Experimental Results

According to the roundness value and profile shape characteristics after centerless grinding, as shown in Table 6, the test specimens are divided into four groups: three-lobe profile with low precision, multiple-lobe profile with low precision, three-lobe profile with high precision, and multiple-lobe profile with high precision. Four pieces were processed in each group, and the profile and roundness value after the experiment were measured. The average value was taken after three measurements for each piece, and the results are shown in Table 8 and Table 9.

The influences of cathode coverage and profile morphology on the variation of roundness error are analyzed below.

#### 3.2.2. Effect of Cathode Coverage on Roundness Improvement

As can be seen from Table 8 and Table 9, when the initial error value is low, the roundness could not be improved with the cathode coverage of 30°. When the cathode coverage is increased to 90°, the roundness error can be further reduced. When the initial error value is high, the roundness can be improved when the cathode coverage is 30 and 90°, as shown in Figure 8. As can be seen from the figure, for the case of low precision, no matter the three-lobe profile or the multiple-lobe profile, the improvement of roundness error when the cathode coverage range is 90° is higher than that when the cathode coverage range is 30°, and for the multiple-lobe profile, the increase of cathode coverage range can achieve a more significant improvement effect. This indicates that an appropriate increase in cathode coverage can further improve the roundness error for both high and low initial errors.

#### 3.2.3. Effect of Initial Roundness Error on Post-Machining Roundness Error

To determine whether there is a significant difference in the effect of the initial roundness error on the roundness error after machining, a one-way ANOVA was performed on the initial roundness and the post-machining roundness when the cathode coverage was 90°, and the results are shown in Table 10 and Table 11. For the three-lobe profile, when the initial roundness is significantly different, the roundness error after machining is also significantly different. For the multiple-lobe profile, when the initial roundness error is significantly different, there is no significant difference in the roundness after machining. This indicates that the final roundness error is not significantly affected by the initial roundness error for the multiple-lobe profile after electrochemical mechanical machining. Therefore, considering the overall machining efficiency, in the centerless grinding stage, it is only necessary to obtain the multiple-lobe profile, and the requirement for the roundness error value can be relaxed appropriately.

#### 3.2.4. Analysis of the Influence of Initial Profile Shape on Post-Processing Accuracy

Based on the analysis results of 3.2.3, the initial roundness error of the multiple-lobe profile does not have a significant effect on the final roundness error at the cathode coverage of 90°. Therefore, the workpieces with different initial errors are considered together to compare the final roundness and the roundness improvement of the three-lobe profile and the multiple-lobe profile, as shown in Figure 9. As can be seen from the figure, the final roundness error of the multiple-lobe profile is lower than that of the three-lobe profile, and the roundness improvement is also better than that of the three-lobe profile.

### 3.3. Experimental Results and Analysis

According to the above experimental results, the roundness error is improved after electrochemical mechanical machining, but the degree of error improvement is different for different profiles, and the high frequency harmonic profile error is easier to be improved. At the same time, increasing the cathode coverage is also beneficial to improve the profile error.

To further investigate the reasons for the differences, the profiles obtained by the two cathode coverage ranges of 30 and 90° were compared, as shown in Table 12 and Table 13. As can be seen from the tables, the profile after electrochemical mechanical machining is related to the original profile. According to the electrochemical mechanical machining theory, when the electrochemical action reaches a stable state, the removal amount of a certain point of the specimen depends on the gap between the point and the cathode, and the change of the gap at the point is mainly affected by the workpiece profile error and process system error. Cathode coverage determines the area on the workpiece surface at the same time where electrochemical effects occur. Therefore, at a certain time, the gap changes caused by process system errors have no difference within the same cathode coverage range, but the gap changes caused by profile errors will affect the removal amount of the workpiece. Figure 10 shows the principle of correcting the error of three-lobe and multiple-lobe profiles. The essential difference between three-lobe and multiple-lobe profiles is that multiple-lobe profiles have high fluctuation frequency, while three-lobe profiles can be regarded as only three fluctuations. At a certain time, the multiple-lobe profiles will have more gap size changes in the whole cathode coverage area, and the instability of the profile error at different machining moments will be homogenized on the machined surface. The more the number of profile lobes, the more obvious this error homogenization effect will be. Therefore, under the same condition, the improvement effect of multiple-lobe profiles is better than that of three-lobe profiles.

According to the above analysis, the gap change caused by the process system error at a certain time has an impact on the entire coverage area of the cathode, and the gap error caused by the process system will spread throughout the coverage area. Therefore, the influence of gap changes caused by process system errors on the removal amount is related to the coverage area. The larger the cathode coverage area is, the more areas on the specimen surface will be processed at the same time, and the instability of the process system error at different processing times will be homogenized in the space of the machined surface, which is conducive to improving the workpiece accuracy.

## 4. Process Conditions Optimization and Processing Cases

### 4.1. Process Conditions Optimization

The optimal configuration principle of process parameters is as follows: when determining process parameters of centerless grinding, the range of process parameters of three-lobe profiles should be avoided. When determining electrochemical mechanical machining process parameters, a larger cathode coverage area should be used, and the limitation of the cathode structure on the coverage area should also be considered. According to the profile shapes under the conditions of different support plate angles and center heights in Figure 5, for the test specimen in this paper, the process parameter combination of centerless grinding that is easy to form multiple-lobe profile should be selected.

The cathode coverage of electrochemical mechanical machining is selected to be near 90°. For other process parameters, select the general process parameters, as shown in Table 14.

### 4.2. Process Conditions Optimization

In order to verify the effectiveness of the optimization range given in this paper, centerless grinding was carried out under the condition that the support plate angle was 70° and the center height was close to 30 mm. Considering the limitation of mechanical structure, the cathode coverage range of 90° was selected for electrochemical mechanical machining. The results are shown in Table 15, Figure 11 and Figure 12. A multiple-lobe profile was obtained by centerless grinding, and after electrochemical mechanical machining, the roundness error value and surface roughness value were greatly reduced at the same time.

## 5. Conclusions

The following main conclusions are obtained according to the experimental study:In a centerless grinding process, increasing the support plate angle can form high-frequency multiple-lobe profiles within a wide range of process parameters.Electrochemical mechanical machining can effectively flatten the high-frequency profile, and appropriately expanding the cathode coverage can also improve the roundness.Under certain cathode coverage, the influence of initial roundness error on final roundness error workpiece is not significant for a multiple-lobe profile.

According to the above conclusions, considering the overall machining efficiency, when the combined process of “centerless grinding + electrochemical mechanical machining” is adopted, to obtain the multiple-lobe profile, we only need to select the optimized process parameters in the centerless grinding stage, and the requirements for the roundness error can be appropriately relaxed, which significantly reduces the process conditions of centerless grinding, and provides an effective machining method for high efficiency and low cost finishing of precision rotary parts. In further research, we will analyze the surface quality of the parts under different parameter configurations of the combined process and adjust the processing parameters under the condition of considering both roughness and accuracy to further optimize the cross-process machining scheme.

## Figures and Tables

**Figure 1 micromachines-14-00063-f001:**
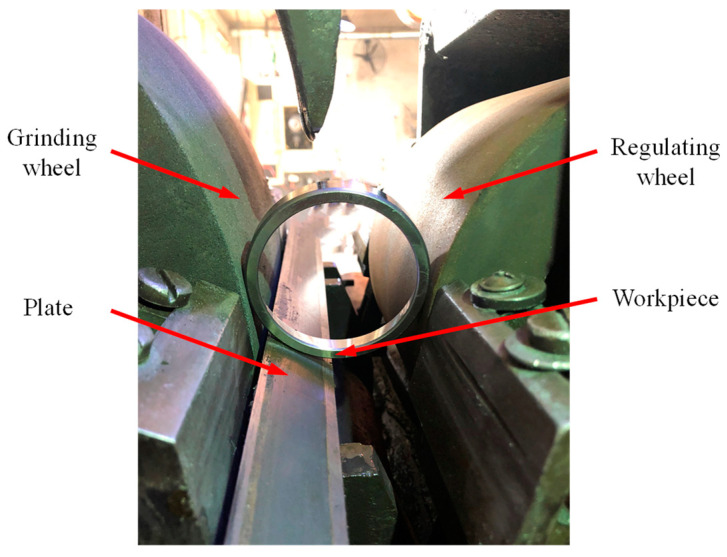
Experimental device.

**Figure 2 micromachines-14-00063-f002:**
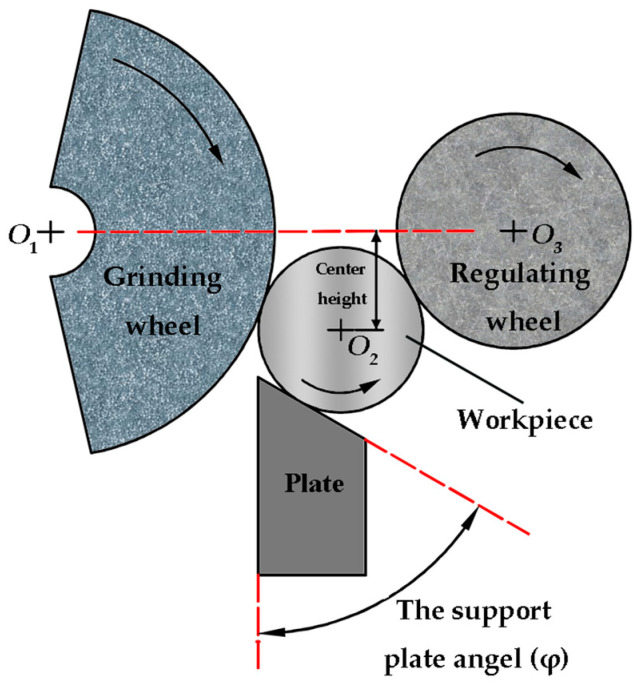
Schematic diagram of center height and the support plate angle.

**Figure 3 micromachines-14-00063-f003:**
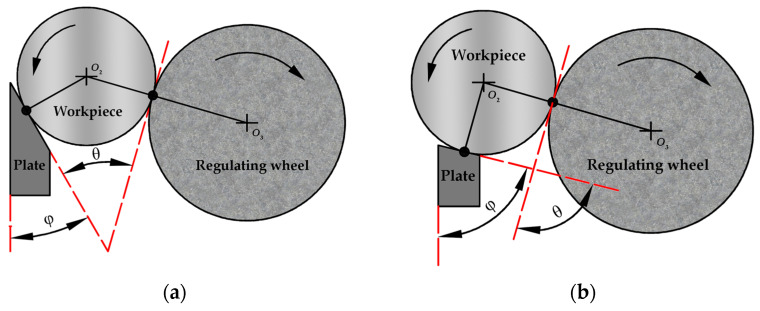
The mechanism of the influence of support plate angle on roundness error: (**a**) the support plate angle is too small; (**b**) the support plate angle is too large.

**Figure 4 micromachines-14-00063-f004:**
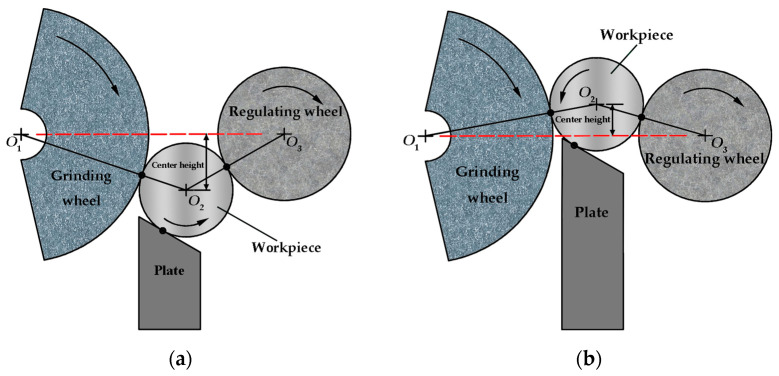
Influence mechanism of center height on profile shape: (**a**) the center height is negative, and the center of the workpiece is lower than the center line of the grinding wheel and the regulating wheel; (**b**) the center height is positive, and the center of the workpiece is higher than the center line of the grinding wheel and the regulating wheel.

**Figure 5 micromachines-14-00063-f005:**
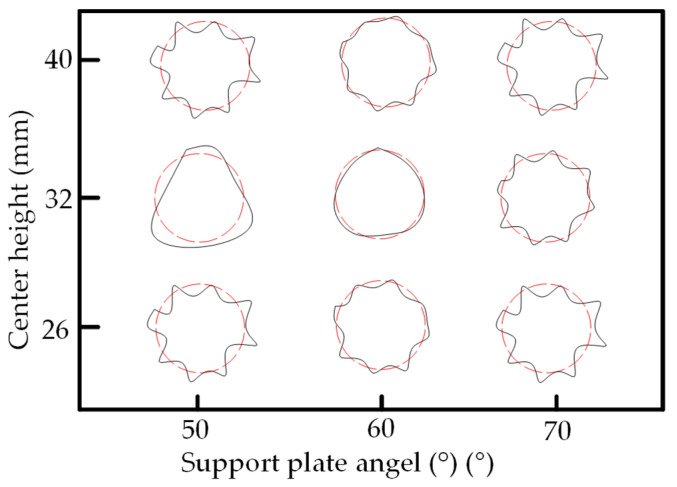
Profile shapes under different processing parameters.

**Figure 6 micromachines-14-00063-f006:**
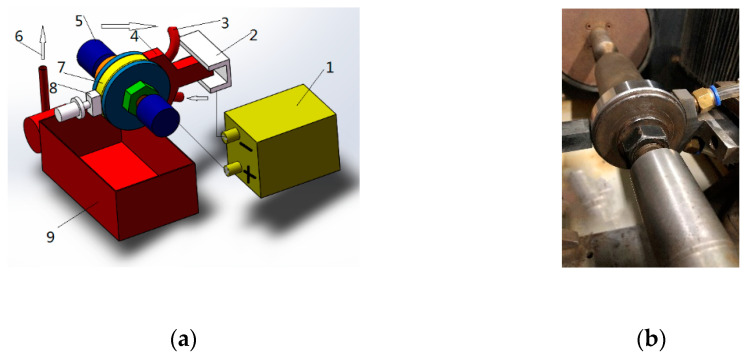
Experimental device: (**a**) schematic diagram, 1—power supply; 2—cathode plate; 3—pipeline; 4—cathode; 5—top; 6—electrolyte; 7—workpiece; 8—grinding tool; 9—electrolyte tank; (**b**) physical diagram.

**Figure 7 micromachines-14-00063-f007:**
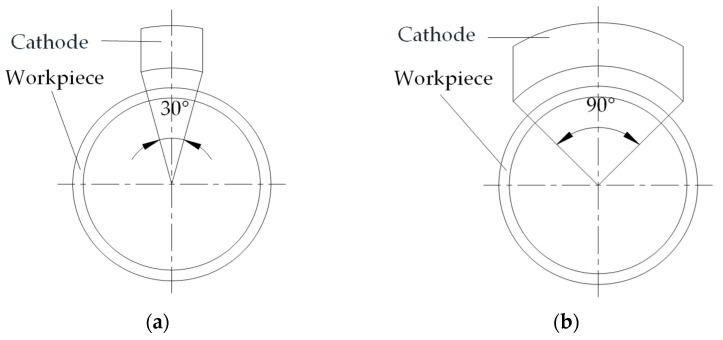
Two kinds of cathode coverage: (**a**) cathode coverage is 30°; (**b**) cathode coverage is 90°.

**Figure 8 micromachines-14-00063-f008:**
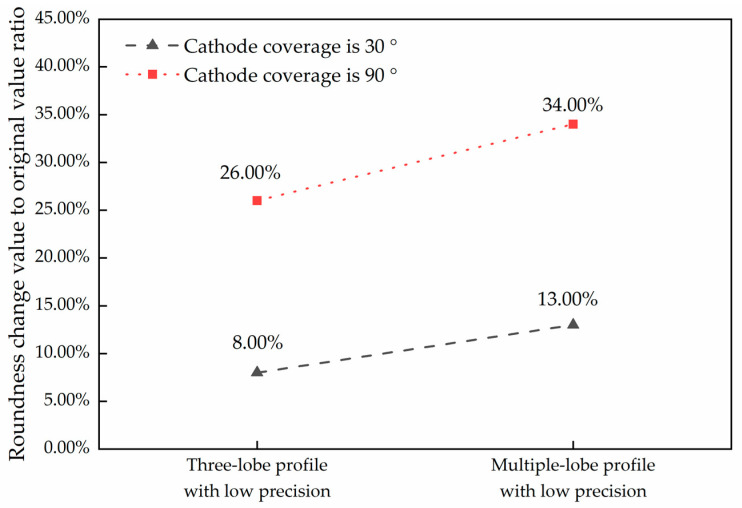
Comparison of roundness improvement after processing of low-precision three-lobe and multiple-lobe profiles.

**Figure 9 micromachines-14-00063-f009:**
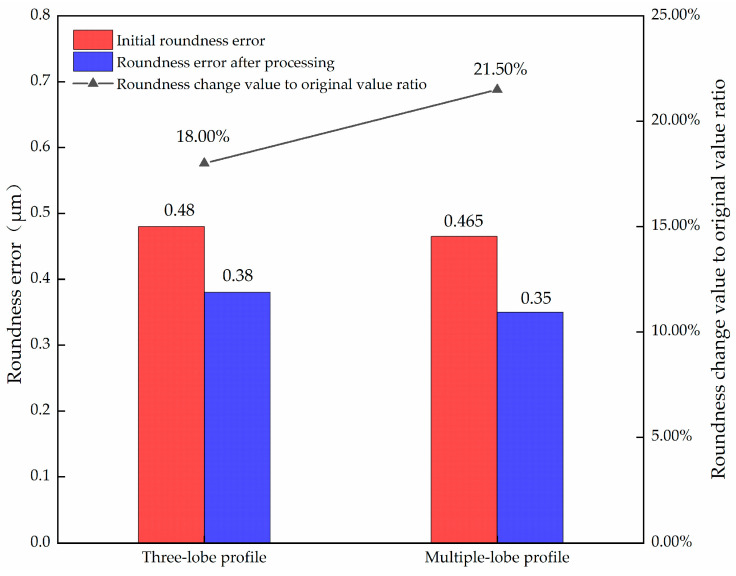
Comparison of roundness after processing between three-lobe profile and multiple-lobe profile.

**Figure 10 micromachines-14-00063-f010:**
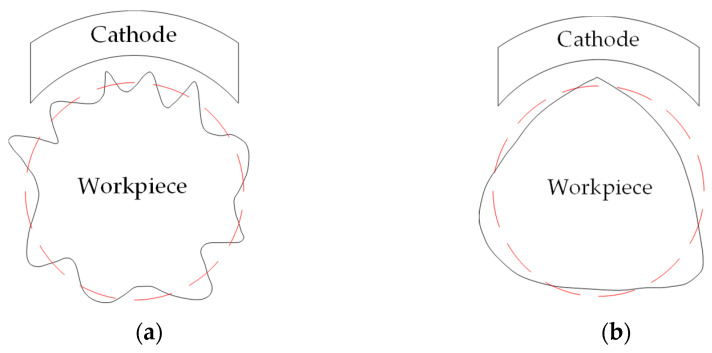
Error correction under different profile shapes: (**a**) multiple-lobe profile; (**b**) three-lobe profile.

**Figure 11 micromachines-14-00063-f011:**
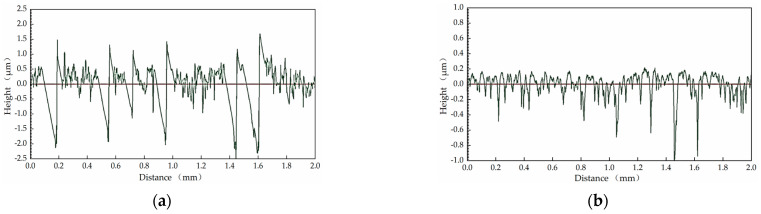
Surface roughness profile of workpiece after machining: (**a**) centerless grinding; (**b**) electrochemical mechanical machining.

**Figure 12 micromachines-14-00063-f012:**
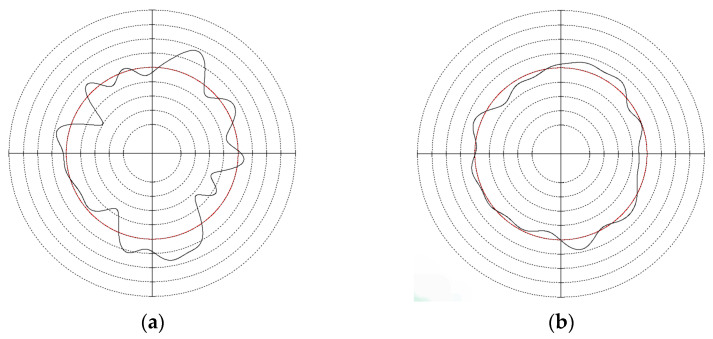
Roundness profile of workpiece after machining: (**a**) centerless grinding; (**b**) electrochemical mechanical machining.

**Table 1 micromachines-14-00063-t001:** Values of centerless grinding experiment parameters.

Support Plate Angle (°)	Center Height (mm)
50	26, 32, 40
60	26, 32, 40
70	26, 32, 40

**Table 2 micromachines-14-00063-t002:** Experimental conditions.

Object	Parameter	Value Range or Model
Experimental equipment	Centerless grinding machine	MK1080
Grinding wheel	Grinding wheel diameter (mm)	500
	Grinding wheel rotation speed (r/min)	1440
	Abrasive materials	White fused alumina
	Particle size (#)	80
Regulating wheel	Regulating wheel diameter (mm)	300
	Regulating wheel speed (r/min)	260
Workpiece	Size (mm)	φ75 × 15
	Material	Bearing steel GCr15
The main parameters	Regulating wheel horizontal inclination angle (°)	0.5
	Radial feed (mm)	0.1
Measuring instrument	Roundness measuring instrument	YS2901

**Table 3 micromachines-14-00063-t003:** Roundness error value obtained under conditions of different center heights and support plate angles.

Center Height (mm)	Support Plate Angle (°)	Workpiece Number										Average Value
		1	2	3	4	5	6	7	8	9	10	
26	50	0.66	0.85	0.76	0.53	0.54	0.48	0.52	0.59	0.53	0.62	0.61
	60	0.61	0.53	0.34	0.3	0.35	0.22	0.23	0.56	0.57	0.34	0.41
	70	0.72	1.28	0.93	0.38	0.51	0.57	0.88	0.69	0.47	0.91	0.73
32	50	0.63	0.53	0.44	0.49	0.42	0.86	0.39	0.54	0.58	0.45	0.53
	60	0.41	0.38	0.32	0.38	0.35	0.28	0.42	0.22	0.4	0.29	0.35
	70	0.42	0.95	0.54	0.54	0.58	0.5	0.65	0.51	0.32	0.43	0.54
40	50	0.59	0.59	0.66	0.48	0.52	0.4	0.6	0.51	0.55	0.65	0.56
	60	0.3	0.62	0.42	0.36	0.59	0.24	0.24	0.35	0.26	0.49	0.39
	70	0.59	0.44	0.45	0.5	0.61	0.76	0.61	0.43	0.64	1.35	0.64

**Table 4 micromachines-14-00063-t004:** Two-way ANOVA with interaction.

Source of Differences	F	*p*-Value	F Crit
Center heights	3.037	0.053	3.109
Support plate angle	18.544	2.34 × 10^−7^	3.109
Interactions	0.458	0.766	2.484

**Table 5 micromachines-14-00063-t005:** Profile shapes before and after processing at different support plate angles and center heights.

Support Plate Angle (°)	Center Height (mm)	Before Processing	After Processing
50	26	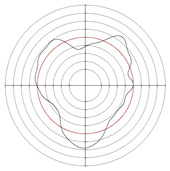	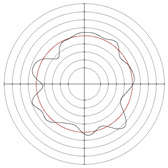
	32	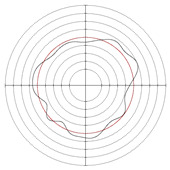	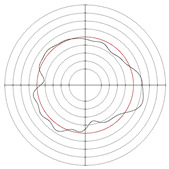
	40	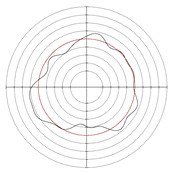	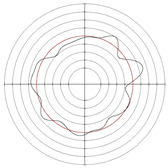
60	26	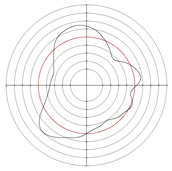	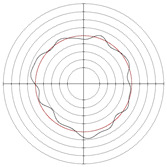
	32	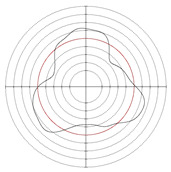	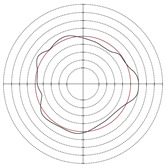
	40	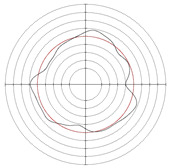	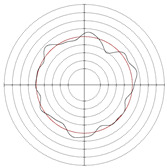
70	26	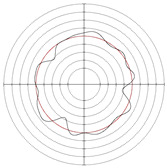	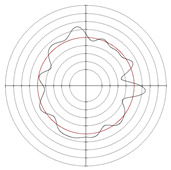
	32	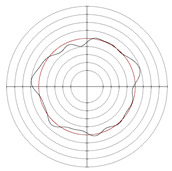	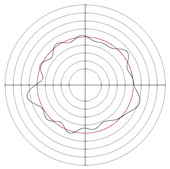
	40	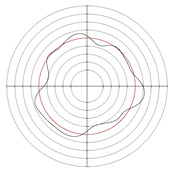	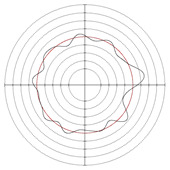

**Table 6 micromachines-14-00063-t006:** Roundness error and profile shape under different processing parameter combinations.

Roundness Error and Profile Shape	Support Plate Angle (°)	Center Height (mm)
Three-lobe profile with low precision	60	32
Multiple-lobe profile with low precision	60	26, 40
Three-lobe profile with high precision	50	32
Multiple-lobe profile with high precision	50	26, 40
	70	26, 32, 40

**Table 7 micromachines-14-00063-t007:** Experimental condition.

Object	Parameters	Range of Value
Cathode	Outlet size (mm)	φ12
	Material	2Cr13
	Surface roughness *Ra* (µm)	1.1~1.3
	cathode coverage (°)	30, 90
Workpiece	Shown in Table 2	Shown in Table 2 (After centerless grinding)
Power supply	Output capacity (KVA)	20
	Input voltage (V)	380
	Output voltage (V)	0~30
	Maximum output current (A)	100
Electrolyte	Flowrate (L/h)	300
	Pressure (MPa)	0.030~0.070
	Main components	NaNO_3_ + Others
	Mass fraction	10~25%
Experimental parameters	Machining current (A)	20
	Machining gap (mm)	0.2
	Revolving speed of workpiece (r/min)	500
	Abrasive pressure (MPa)	0.1
	Machining time (s)	30
	Abrasive tool	Abrasive belt (30#)
Measuring device	Surface roughness tester	YS2205B
	Roundness measuring instrument	YS2901
	Flow meter	LZS-15
	pressure gauge	Y-60

**Table 8 micromachines-14-00063-t008:** Experimental results with 30° cathode coverage.

Type	Workpiece	Roundness before Processing (µm)	Roundness after Processing (µm)	Roundness Change Value to Original Value Ratio (Keep Two Valid Digits)
Three-lobe profile with low precision	1	0.85	0.75	0.12
	2	0.51	0.59	−0.16
	3	0.85	0.77	0.10
	4	0.60	0.44	0.27
	Average value	0.70	0.64	0.08
Multiple-lobe profile with low precision	1	0.50	0.56	−0.11
	2	0.57	0.46	0.20
	3	1.35	0.55	0.60
	4	0.48	0.56	−0.17
	Average value	0.73	0.53	0.13
Three-lobe profile with high precision	1	0.38	0.40	−0.04
	2	0.41	0.33	0.20
	3	0.40	0.48	−0.19
	4	0.38	0.63	−0.64
	Average value	0.39	0.46	−0.17
Multiple-lobe profile with high precision	1	0.35	0.47	−0.33
	2	0.46	0.45	0.03
	3	0.32	0.50	−0.55
	4	0.43	0.64	−0.49
	Average value	0.39	0.51	−0.33

**Table 9 micromachines-14-00063-t009:** Experimental results with 90° cathode coverage.

Type	Workpiece	Roundness before Processing (µm)	Roundness after Processing (µm)	Roundness Change Value to Original Value Ratio (Keep Two Valid Digits)
Three-lobe profile with low precision	1	0.72	0.44	0.39
	2	0.68	0.48	0.29
	3	0.61	0.43	0.30
	4	0.51	0.49	0.05
	Average value	0.63	0.46	0.26
Multiple-lobe profile with low precision	1	0.53	0.37	0.31
	2	0.63	0.48	0.25
	3	0.46	0.30	0.35
	4	0.65	0.35	0.46
	Average value	0.57	0.37	0.34
Three-lobe profile with high precision	1	0.34	0.35	−0.02
	2	0.34	0.25	0.26
	3	0.30	0.28	0.07
	4	0.35	0.33	0.07
	Average value	0.33	0.3	0.1
Multiple-lobe profile with high precision	1	0.34	0.32	0.06
	2	0.30	0.30	0.02
	3	0.38	0.37	0.03
	4	0.43	0.32	0.26
	Average value	0.36	0.33	0.09

**Table 10 micromachines-14-00063-t010:** One factor ANOVA of roundness of workpiece with the three-lobe profile.

Roundness	F	*p*-Value	F Crit
Original	39.51907	0.000754	5.987378
Processed	33.54085	0.00116	5.987378

**Table 11 micromachines-14-00063-t011:** One factor ANOVA of roundness of workpiece with the multiple-lobe profile.

Roundness	F	*p*-Value	F Crit
Original	15.30501	0.007873	5.987378
Processed	1.355444	0.288524	5.987378

**Table 12 micromachines-14-00063-t012:** Profile shape before and after processing under 30° cathode coverage.

Workpiece Profile	Type	Profile Shape			
Before processing	Three-lobe profile with low precision	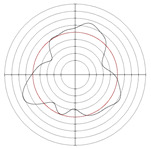	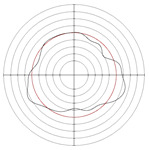	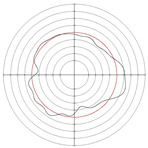	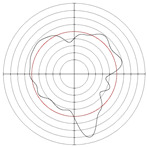
After processing		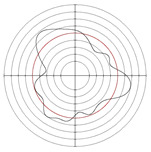	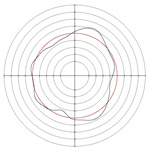	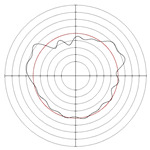	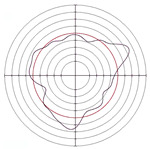
Before processing	Multiple-lobe profile with low precision	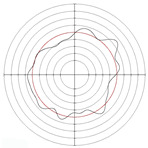	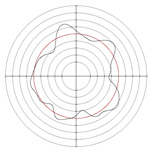	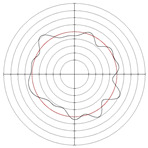	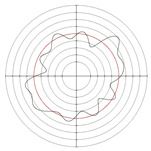
After processing		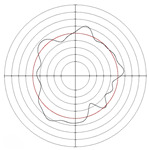	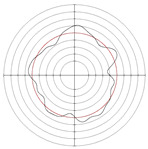	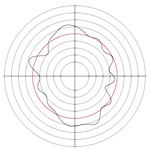	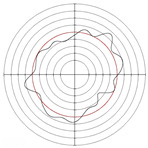

**Table 13 micromachines-14-00063-t013:** Profile shape before and after processing under 90° cathode coverage.

Workpiece Profile	Type	Profile Shape			
Before processing	Three-lobe profile with low precision	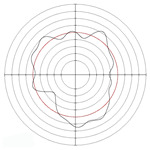	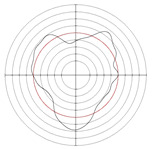	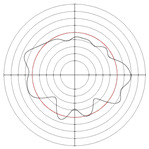	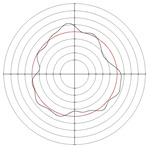
After processing		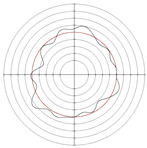	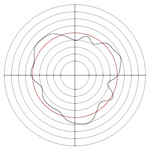	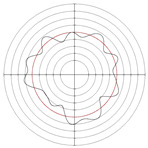	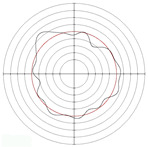
Before processing	Multiple-lobe profile with low precision	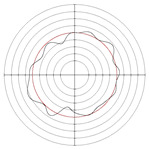	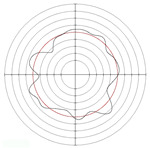	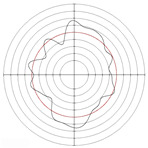	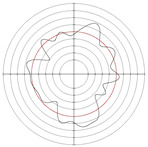
After processing		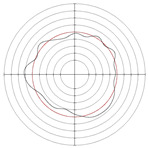	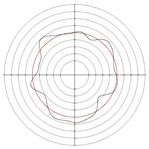	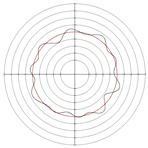	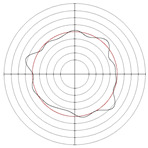

**Table 14 micromachines-14-00063-t014:** Processing conditions.

Technology	Parameters	Range of Value
Centerless grinding	Grinding wheel material	White fused alumina
	Grinding wheel granularity (#)	80
	Speed of grinding wheel (r/min)	1440
	Regulating wheel speed (r/min)	260
	Regulating wheel horizontal inclination (°)	0.5
	Radial feed (mm)	0.1
	Processing time (s)	20
Electrochemical mechanical machining	Machining current (A)	20
	Machining gap (mm)	0.2
	Revolving speed of workpiece (r/min)	500
	Abrasive pressure (MPa)	0.1
	Machining time (s)	30
	Abrasive tool	Abrasive belt (30#)

**Table 15 micromachines-14-00063-t015:** Experimental results after centerless grinding and electrochemical mechanical machining.

Workpiece	Surface Roughness *Ra* after Centerless Grinding (μm)	Surface Roughness *Ra* after Electrochemical Mechanical Machining (μm)	Roundness after Centerless Grinding (μm)	Roundness after Electrochemical Mechanical Machining (μm)
1	0.488	0.094	0.69	0.39
2	0.567	0.133	0.56	0.38
3	0.442	0.147	0.63	0.31
4	0.460	0.119	0.49	0.37
Average value	0.489	0.123	0.592	0.362

## Data Availability

Not applicable.

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
