# Peer review of "Profile Evolution and Cross-Process Collaboration Strategy of Bearing Raceway by Centerless Grinding and Electrochemical Mechanical Machining"

_micromachines, 2022, doi:10.3390/mi14010063_

Round 1

Reviewer 1 Report

The combined machining technology of "centerless grinding + electrochemical mechanical machining" provides an efficient technical means to realize the precision finishing machining of rotary parts such as bearing raceways.  The processing method proposed by the author is feasible and can improve roundness well. The theory is right, the experiment is complete. However, the Fig 7, Fig 8 and Fig 10 is not very clear and I can not fully see the data and interpretation in the figure. 

1.    What is the main question addressed by the research?

According to the experimental results, the combination of the two techniques "centerless grinding + electrochemical mechanical machining" has a good effect on improving roundness.

2.    Do you consider the topic original or relevant in the field? Does it address a specific gap in the field?

Relevant.

In centerless grinding process, the electrochemical mechanical machining improve the roundness.

3.    What does it add to the subject area compared with other published material?

"centerless grinding + electrochemical mechanical machining"

4.    What specific improvements should the authors consider regarding the methodology? What further controls should be considered?

In Table 1, particle size of abrasive, rotational speed of grinding wheel and regulating wheel, etc., why the author chose these as experimental parameters and whether they have been optimized by the author after previous experiments?

5.    Are the conclusions consistent with the evidence and arguments presented and do they address the main question posed?

Yes.

6.    Are the references appropriate?

Yes.

      7. Please include any additional comments on the tables and figures.

The graphics are not clear, especially in Figure 7,8 and 10.

  •  

Reviewer 2 Report

This paper explores the influence of process parameters on the roundness error from the perspective of profile evolution during centerless grinding and electrochemical mechanical machining, and puts forward a combined machining technology of "centerless grinding + electrochemical mechanical machining", which provides an efficient technical means to realize the precision finishing machining of rotary parts such as bearing raceways. The research is innovative and has good practical application value. The paper has a complete experimental process and detailed data, as well as in-depth theoretical analysis and rigorous demonstration. The work is scientifically suitable for publication.

A few suggestions are as follows :

1. The meanings of the support plate angel and the center height should be illustrated in the experimental design part of centerless grinding.

2. In 1.2.2, the discussion on the influence law of center height on contour shape should be clearer. Combined with the following analysis, it should be expressed as” in a certain range of the support plate angel, the center height has a significant effect on the profile shape, and increasing or lowering the center height will make the multiple-lobe profile.”

3. In Figure 4, Pallet apex angle should be changed to “support plate angel” to be consistent with the text.

4. The variation law of roughness can be further discussed for the proposed cross-process collaboration strategy,and I hope that this problem will be discussed in the next article.
